# Down-regulation of TORC2-Ypk1 signaling promotes MAPK-independent survival under hyperosmotic stress

Alexander Muir[1,2†], Françoise M Roelants[1], Garrett Timmons[3], Kristin L Leskoske[1], Jeremy Thorner[1]*

[1]Division of Biochemistry, Biophysics and Structural Biology, Department of Molecular and Cell Biology, University of California, Berkeley, Berkeley, United States; [2]Chemical Biology Graduate Program, University of California, Berkeley, Berkeley, United States; [3]Department of Chemistry, University of California, Berkeley, Berkeley, United States

**Abstract** In eukaryotes, exposure to hypertonic conditions activates a MAPK (Hog1 in *Saccharomyces cerevisiae* and ortholog p38 in human cells). In yeast, intracellular glycerol accumulates to counterbalance the high external osmolarity. To prevent glycerol efflux, Hog1 action impedes the function of the aquaglyceroporin Fps1, in part, by displacing channel co-activators (Rgc1/2). However, Fps1 closes upon hyperosmotic shock even in *hog1Δ* cells, indicating another mechanism to prevent Fps1-mediated glycerol efflux. In our prior proteome-wide screen, Fps1 was identified as a target of TORC2-dependent protein kinase Ypk1 (*Muir et al., 2014*). We show here that Fps1 is an authentic Ypk1 substrate and that the open channel state of Fps1 requires phosphorylation by Ypk1. Moreover, hyperosmotic conditions block TORC2-dependent Ypk1-mediated Fps1 phosphorylation, causing channel closure, glycerol accumulation, and enhanced survival under hyperosmotic stress. These events are all Hog1-independent. Our findings define the underlying molecular basis of a new mechanism for responding to hypertonic conditions.

*For correspondence: jthorner@berkeley.edu

Present address: †Vander Heiden Lab, Department of Biology and Koch Institute for Integrative Cancer Research, Massachusetts Institute of Technology, Cambridge, United States

Competing interests: The authors declare that no competing interests exist.

## Introduction

In *Saccharomyces cerevisiae*, target of rapamycin (TOR) Complex 2 (TORC2)-dependent signaling responds to multiple plasma membrane-perturbing stresses, including sphingolipid depletion (*Roelants et al., 2010, 2011*), heat shock (*Sun et al., 2012*), and both hypotonic (*Berchtold et al., 2012*) and hypertonic (*Lee et al., 2012*) conditions. The essential downstream effector of TORC2 is the protein kinase Ypk1 (and its paralog Ypk2) (*Casamayor et al., 1999*; *Roelants et al., 2002, 2004*).

To understand how TORC2-Ypk1 signaling elicits cellular responses, we performed a genome-wide screen to discern previously unidentified Ypk1 substrates and thereby discovered that ceramide synthase activity is stimulated by TORC2-Ypk1 (*Muir et al., 2014*). Among other potential targets, our screen pinpointed two proteins involved in glycerol metabolism, aquaglyceroporin Fps1 (*Luyten et al., 1995*) and Gpt2/Gat1, an enzyme that converts glycerol-3P to phosphatidic acid (*Zheng and Zou, 2001*), a precursor to other phospholipids and triacylglycerol (*Henry et al., 2012*). Both candidates warranted further investigation because, as we showed, Ypk1-mediated phosphorylation inhibits another enzyme in glycerol metabolism, Gpd1, which generates glycerol-3P (*Lee et al., 2012*), and *GPD1* expression is highly induced by hyperosmotic stress (*Albertyn et al., 1994b*). Second, accumulation of intercellular glycerol is essential for yeast cell survival under hyperosmotic conditions (*Westfall et al., 2008*; *Saito and Posas, 2012*; *Hohmann, 2015*). Thus, we suspected that TORC2-Ypk1 signaling might play an as yet unrecognized role in the cellular response to hyperosmotic shock.

Hyperosmotic conditions evoke two known signaling modalities. Pathways coupled to alternative osmosensors (Sln1 and Sho1) activate MAPK Hog1, which drives both transcription-independent and -dependent responses that markedly increase both production and intracellular retention of glycerol (*Westfall et al., 2008*; *Saito and Posas, 2012*; *Hohmann, 2015*). Hyperosmotic shock also increases cytosolic [$Ca^{2+}$] thereby activating calcineurin (CN) (*Denis and Cyert, 2002*), promoting processes that stimulate retrieval of excess plasma membrane (*Guiney et al., 2015*). Although both CN-deficient and *hog1Δ* cells are quite sensitive to the ionic imbalances caused by high salt (e.g., 1 M NaCl), *hog1Δ* cells are significantly more sensitive to hypertonic stress per se, such as a high concentration of an uncharged impermeant osmolyte (e.g., 1 M sorbitol).

Our understanding of the response to high osmolarity remains incomplete, however. Although it is well documented that preventing glycerol efflux through the aquaglyceroporin Fps1 is essential for yeast to survive hyperosmolarity (*Luyten et al., 1995*; *Tamás et al., 1999*; *Duskova et al., 2015*), and that activated Hog1 can negatively regulate this channel by displacing the Fps1-activating proteins Rgc1/2 (*Lee et al., 2013*), Fps1 still closes in response to hyperosmotic shock in *hog1Δ* cells (*Tamás et al., 1999*; *Babazadeh et al., 2014*). Therefore, we explored the possibility, as suggested by our screen, that Fps1 is an authentic target of TORC2-dependent Ypk1-mediated phosphorylation, that this modification is important for Fps1 function, and that it is under regulation by hyperosmotic conditions.

## Results

### Ypk1 phosphorylates Fps1 and hyperosmotic shock inhibits this phosphorylation

The 743-residue enzyme Gpt2 contains one Ypk1 phospho-acceptor motif ($_{646}$**R**S**R**SS**SI**$_{652}$). At such sites, Ser residues just penultimate to the canonical one (in red) can be phosphorylated in a Ypk1-dependent manner (*Roelants et al., 2011*). Therefore, we generated a Gpt2(S649A S650A S651A) mutant. One or more of these three Ser residues is phosphorylated in vivo because, compared to wild-type, Gpt2$^{3A}$ exhibited a distinctly faster mobility upon SDS-PAGE, a hallmark of decreased phosphorylation (*Figure 1A*), just like wild-type Gpt2 treated with phosphatase (*Figure 1—figure supplement 1*). However, this phosphorylation did not appear to be dependent on Ypk1 because little change occurred in Gpt2 mobility when an analog-sensitive *ypk1-as ypk2Δ* strain was treated with the cognate inhibitor (3-MB-PP1) (*Figure 1A*).

In marked contrast, three of four predicted Ypk1 sites in the 669-residue Fps1 channel ($_{176}$**RRR**S**R**S**R**$_{182}$, $_{180}$**R**SRAT**S**N$_{186}$, $_{565}$**R**A**R**RT**S**D$_{571}$) (*Figure 1—figure supplement 2A*) are phosphorylated in vivo, as indicated by the effect of site-directed mutations to Ala on electrophoretic mobility (*Figure 1—figure supplement 2B*), and their phosphorylation requires Ypk1 activity, because, in inhibitor-treated *ypk1-as ypk2Δ* cells, the mobility of wild-type Fps1 was indistinguishable from that of Fps1(S181A S185A S570A) (*Figure 1B*), just like wild-type Fps1 treated with phosphatase (*Figure 1—figure supplement 2C*). Moreover, a C-terminal fragment of Fps1 containing Ser570, one of the apparent Ypk1 phosphorylation sites delineated in vivo, is phosphorylated by purified Ypk1 in vitro and solely at the Ypk1 site (S570) (*Figure 1—figure supplement 3*). Furthermore, as for other Ypk1-dependent modifications (*Muir et al., 2014*), phosphorylation of these same sites in Fps1 in vivo was also TORC2-dependent, because treatment with a TORC2 inhibitor (NVP-BEZ235) (*Kliegman et al., 2013*) also reduced Fps1 phosphorylation (*Figure 1C*). Thus, Fps1 is a *bona fide* Ypk1 substrate.

We documented elsewhere using Phos-tag gel mobility shift that Ypk1 phosphorylation at T662, one of its well-characterized TORC2 sites, is eliminated when cells are subjected to hyperosmotic shock for 10 min (*Lee et al., 2012*), and the same effect is observed using a specific antibody (*Niles et al., 2012*) that monitors phosphorylation of Ypk1 at the same site (*Figure 1—figure supplement 4A*). Using Ypk1$^{7A}$, which also permits facile detection of mobility shifts arising from TORC2-specific phosphorylation (K Leskoske and FM Roelants, unpublished results) (*Figure 1—figure supplement 4B*), we followed the kinetics of this change. Loss of TORC2-mediated Ypk1 phosphorylation upon hyperosmotic shock occurs very rapidly (within 1 min) and persists for about 15 min (*Figure 1D*), but is transient. By 20 min after hyperosmotic shock, TORC2-mediated Ypk1 phosphorylation is again detectable and is nearly back to the pre-stress level by 75 min (*Figure 1—figure supplement 5A*). Rapid reduction in TORC2-mediated Ypk1 phosphorylation under hypertonic stress was still observed in mutants lacking the Sho1- or Sln1-dependent pathways that converge on Hog1 or Hog1

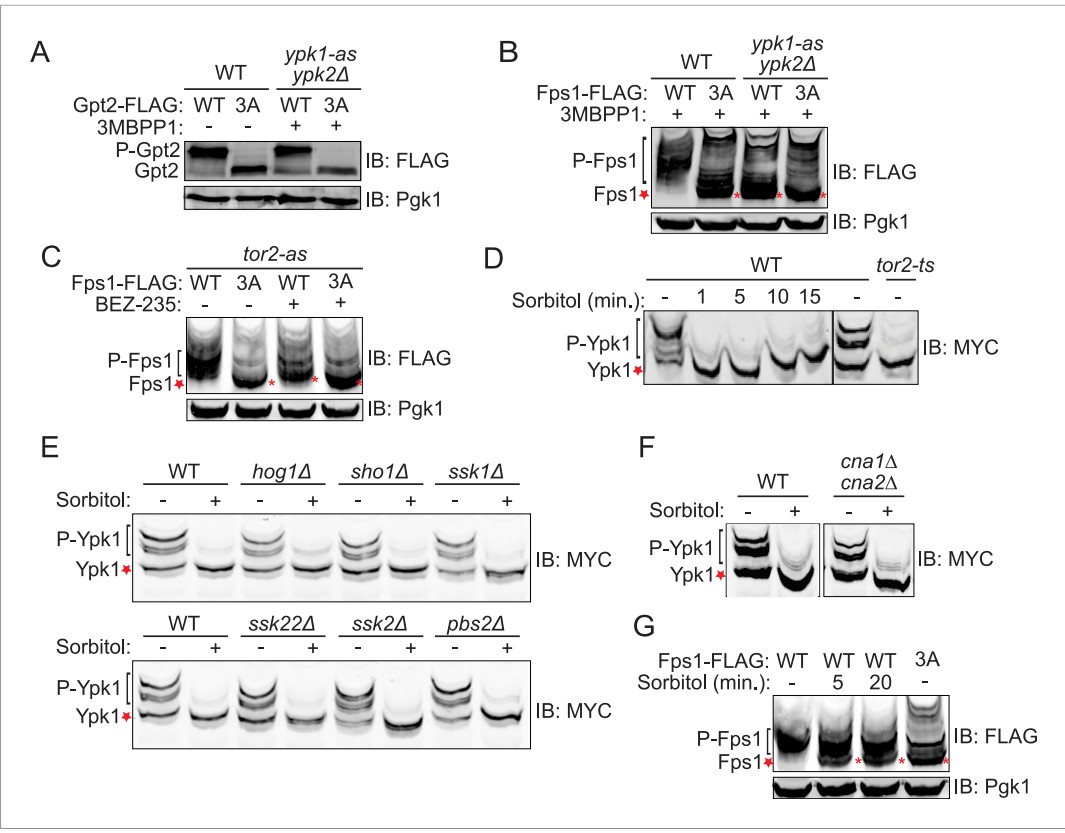

**Figure 1**. Fps1 (but not Gpt2) is phosphorylated by Ypk1. (**A**) Wild-type (BY4741) or *ypk1-as ypk2Δ* (yAM135-A) cells expressing plasmid borne Gpt2-3xFLAG (pAX238) or Gpt2³ᴬ-3xFLAG (pAX244) were grown to mid-exponential phase and then treated with vehicle (−) or 10 μM 3-MB-PP1 (+) for 90 min. Cells were harvested, extracts prepared, resolved by SDS-PAGE, and blotted as in 'Materials and methods'. (**B**) Wild-type cells expressing either Fps1-3xFLAG (yGT21) or Fps1³ᴬ-3xFLAG (yGT22) from the *FPS1* promoter at the normal chromosomal locus, or *ypk1-as ypk2Δ* cells expressing either Fps1-3xFLAG (yAM281) or Fps1³ᴬ-3xFLAG (yAM284-A) from the *FPS1* promoter at the normal chromosomal locus, were grown to mid-exponential phase and treated as in (**A**) with vehicle or 3-MB-PP1 for 60 min. Cells were harvested, extracts prepared, resolved by Phos-tag SDS-PAGE, and blotted as in 'Materials and methods'. Unphosphorylated Fps1 (red asterisk). (**C**) A *tor2-as* strain (yKL5) expressing Fps1-3xFLAG (pAX274) or Fps1³ᴬ-3xFLAG (pAX275) was grown to mid-exponential phase and then treated with vehicle (−) or 2 μM BEZ-235 (+) for 30 min. Cells were harvested, extracts prepared, resolved and analyzed as in (**B**). (**D**) Wild-type (BY4741) or *tor2-29ᵗˢ* (JTY5468) cells expressing Ypk1⁷ᴬ-myc (pFR252) were grown at 30°C (*left panel*) or 26°C (*right panel*) to mid-exponential phase, then diluted into fresh YPD in the absence (−) or presence of 1 M sorbitol (final concentration). After the indicated times (1–15 min), culture samples were collected, lysed and the resulting extracts resolved by Phos-tag SDS-PAGE and analyzed by immunoblotting with anti-myc mAb 9E10, as described in 'Materials and methods'. (**E**) As in (**D**), except for the genotype (strain) expressing Ypk1⁷ᴬ-myc (pFR252), which were, aside from the wild-type control, *hog1Δ* (YJP544), *sho1Δ* (JTY5540), *ssk1Δ* (JTY5541), *ssk22Δ* (JTY5539), *ssk2Δ* (JTY5538) or *pbs2Δ* (JTY5537), and the treatment with 1 M sorbitol was for 1 min. (**F**) Wild-type (BY4741) or otherwise isogenic *cna1Δ cna2Δ* (JTY5574) cells expressing Ypk1⁷ᴬ-myc (pFR252) were grown to mid-exponential phase then diluted into fresh YPD in the absence (−) or presence (+) of 1 M sorbitol (final concentration). After 1 min, the cells were collected, lysed and the resulting extracts resolved by Phos-tag SDS-PAGE and analyzed by immunoblotting with anti-myc mAb 9E10, as described in 'Materials and methods'. (**G**) Wild-type cells expressing either Fps1-3xFLAG (yGT21) or Fps1³ᴬ-3xFLAG (yGT22) from the chromosomal *FPS1* locus, were diluted into fresh YPD in the absence (−) or presence of 1 M sorbitol (final concentration) for the indicated times and then extracts of the cells prepared and analyzed as in (**B**).

The following figure supplements are available for figure 1:

**Figure supplement 1**. Gpt2 is a phosphoprotein in vivo.

**Figure supplement 2**. Fps1 is phosphorylated at three predicted Ypk1 sites in vivo.

*Figure 1. continued on next page*

*Figure 1. Continued*

**Figure supplement 3**. A fragment carrying one of the in vivo Ypk1-dependent sites in Fps1 is phosphorylated by purified Ypk1 in vitro exclusively on the same site.

**Figure supplement 4**. Modification at T662 and isoforms of Ypk1[7A] both accurately report authentic in vivo phosphorylation.

**Figure supplement 5**. Hyperosmotic shock induced loss of Ypk1 and Fps1 phosphorylation is transient.

itself (*Figure 1E*) or CN (*Figure 1F*). Thus, loss of TORC2-mediated Ypk1 phosphorylation upon hyperosmotic shock occurs independently of other known response pathways.

Given that Ypk1 phosphorylates Fps1 and that hyperosmotic stress rapidly abrogates TORC2-dependent phosphorylation and activation of Ypk1, Ypk1 modification of Fps1 should be prevented under hyperosmotic stress. As expected, Ypk1 phosphorylation of Fps1 is rapidly lost upon hyperosmotic shock (*Figure 1G*), yielding a species with mobility indistinguishable from Fps1[3A], remains low for at least 20 min, but returns by 75 min (*Figure 1—figure supplement 5B*), mirroring the kinetics of loss and return of both TORC2-mediated Ypk1 phosphorylation (*Figure 1D* and *Figure 1—figure supplement 5A*) and Ypk1-dependent phosphorylation of Gpd1 that we observed before (*Lee et al., 2012*). Thus, hyperosmotic stress dramatically down-modulates Ypk1-mediated phosphorylation of Fps1.

## Ypk1 phosphorylation of Fps1 promotes channel opening and glycerol efflux

In its open state, the Fps1 channel permits entry of toxic metalloid, arsenite, which inhibits growth (*Thorsen et al., 2006*), whereas lack of Fps1 (*fps1Δ*) or the lack of channel activators (*rgc1Δ rgc2Δ*) (*Beese et al., 2009*) or an Fps1 mutant that cannot open because it cannot bind the activators (Fps1[ΔPHD]) (*Lee et al., 2013*) are arsenite resistant. We found that Fps1[3A] was at least as arsenite resistant as any other mutant that abrogates Fps1 function (*Figure 2A*). Thus, Fps1[3A] acts like a closed channel, suggesting that Ypk1-mediated phosphorylation promotes channel opening. Loss of individual phosphorylation sites led to intermediate levels of arsenite resistance (*Figure 2B*). Thus, modification at these sites contributes additively to channel opening.

Others have shown that intracellular glycerol is elevated in *fps1Δ* cells in the absence of hyperosmotic stress (*Tamás et al., 1999*). If Fps1[3A] favors the closed-channel state, then it should also cause constitutive elevation of intracellular glycerol concentration. Indeed, in the absence of any osmotic perturbation, Fps1[3A] mutant cells accumulated ~twofold as much glycerol as otherwise isogenic *FPS1⁺* strains (*Figure 2C*). Consistent with this result, we observed before that loss of Ypk1 (and Ypk2) activity caused an increase in glycerol level compared to control cells (*Lee et al., 2012*).

Consistent with Ypk1-dependent phosphorylation affecting Fps1 channel function per se, immunoblotting (*Figure 2D*) and fluorescence microscopy (*Figure 2E*) showed that the steady-state level and localization of Fps1 are unaffected by the presence or absence of these modifications.

## Hyperosmotic stress-evoked down-regulation of Ypk1 phosphorylation of Fps1 promotes cell survival independently of known Fps1 regulators

Fps1 can be negatively regulated by Hog1 via two mechanisms: Hog1 phosphorylation of Fps1 stimulates its internalization and degradation (*Thorsen et al., 2006*; *Mollapour and Piper, 2007*); Hog1 phosphorylation closes the channel by displacing bound Fps1 activators (Rgc1 and Rgc2) (*Beese et al., 2009*; *Lee et al., 2013*). We found, however, that Fps1[3A] was still in the closed state, as judged by arsenite resistance, in the total absence of Hog1 (*hog1Δ*) (*Figure 3A*), or in an Fps1 mutant (Fps1[IVAA]) that cannot bind Hog1 or where the activator cannot be displaced from Fps1 by Hog1 phosphorylation (Rgc2[7A]) (*Lee et al., 2013*) (*Figure 3B*). Thus, closure of the Fps1 channel by lack of Ypk1 phosphorylation occurs independently of any effects requiring Hog1. Consistent with this conclusion, presence or absence of Ypk1-mediated Fps1 phosphorylation had no effect on Fps1-Rgc2 interaction (*Figure 3C*).

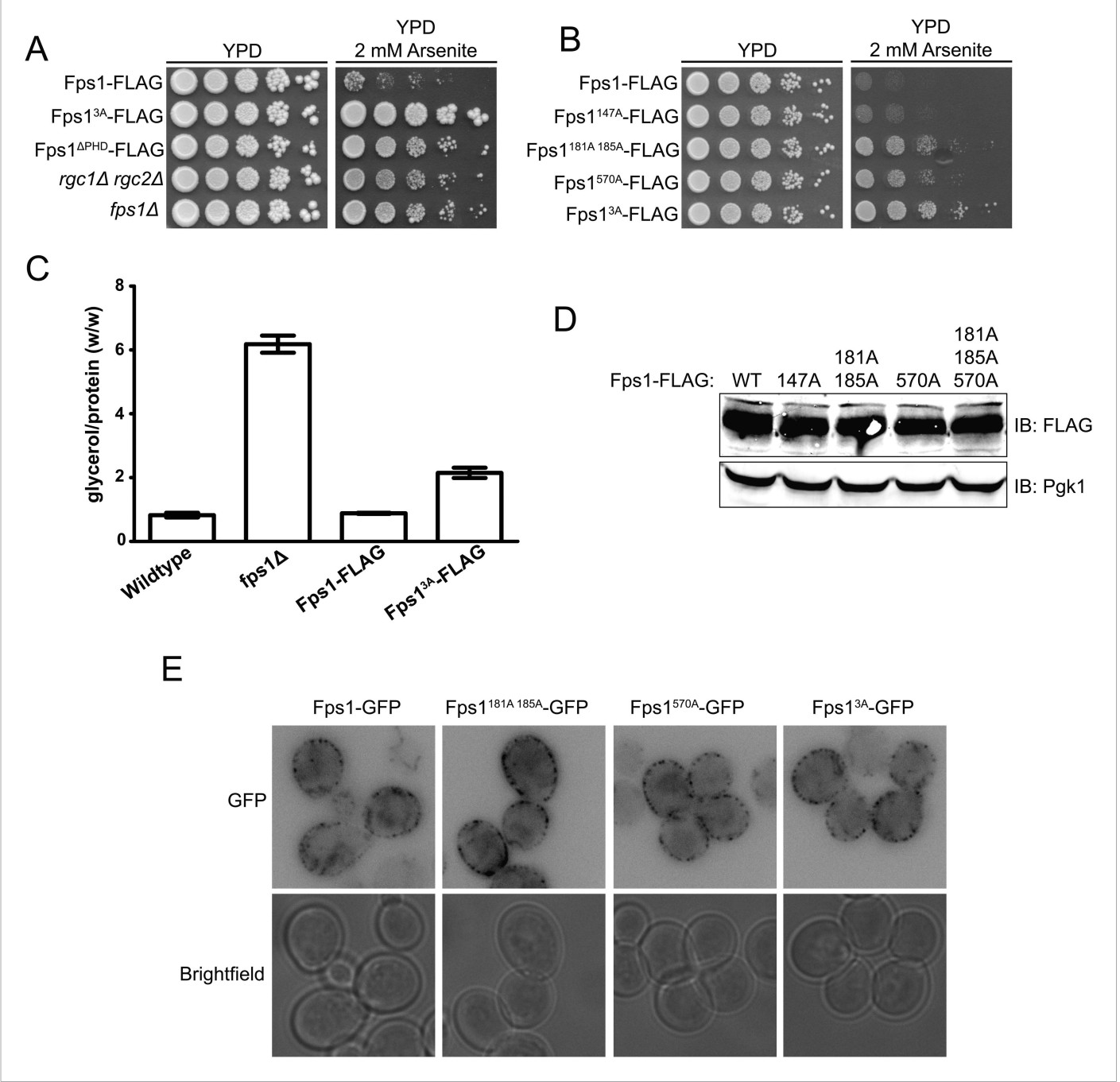

**Figure 2**. Phosphorylation by Ypk1 opens the Fps1 channel. (**A**) Cultures of Fps1-3xFLAG (yGT21), Fps1³ᴬ-3xFLAG (yGT22), Fps1^ΔPHD-3xFLAG (yAM307-A), *rgc1Δ rgc2Δ* (DL3188) and *fps1Δ* (yAM181-A) were adjusted to $A_{600\,nm} = 1.0$ and serial dilutions were then spotted onto YPD plates containing the indicated concentration of arsenite. Cells were allowed to grow for 4 days at 30°C prior to imaging. (**B**) As in (**A**), except Fps1-3xFLAG (yGT21), Fps1 (T147A)-3xFLAG (yAM310-A), Fps1(S181A S185A)-3xFLAG (yAM301-A), Fps1(S570A)-3xFLAG (yGT24) or Fps1³ᴬ-3xFLAG (yGT22) cultures were used and cells were grown for 2 days at 30°C prior to imaging. (**C**) Triplicate exponentially-growing cultures of wild-type (BY4742), *fps1Δ* (yAM181-A), Fps1-3xFLAG (yGT21) and Fps1³ᴬ-3xFLAG (yGT22) strains were harvested, extracted, and the glycerol and protein concentration measured as described in 'Materials and methods'. Values represent the ratio of glycerol-to-protein (error bar, standard error of the mean). (**D**) Extracts from the strains in (**B**) were resolved by standard SDS-PAGE using 8% acrylamide gels. (**E**) *fps1Δ* (yAM181-A) cells expressing Fps1-GFP (pAX290), Fps1(S181A S185A)-GFP, (pAX294), Fps1 (S570A)-GFP (pAX293) or Fps1³ᴬ-GFP (pAX295) were viewed by fluorescence microscopy as described in 'Materials and methods'. Representative fields are shown.

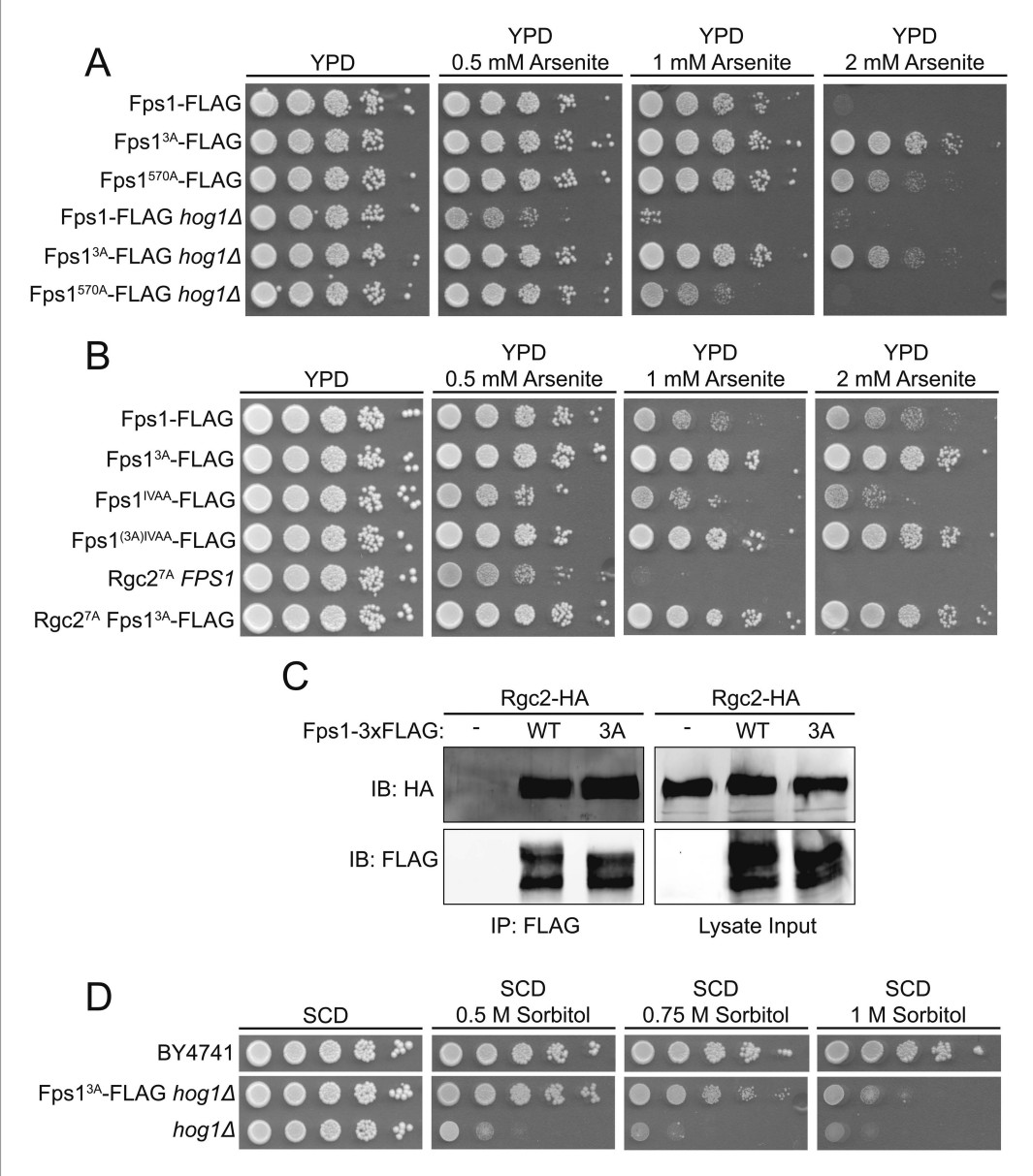

**Figure 3**. TOR Complex 2 (TORC2)-dependent Ypk1-mediated regulation of Fps1 is independent of Hog1 and Rgc1 and Rgc2. (**A**) Cultures of Fps1-3xFLAG (yGT21), Fps1$^{570A}$-3xFLAG (yGT24), Fps1$^{3A}$-3xFLAG (yGT22), Fps1-3xFLAG *hog1Δ* (yAM275), Fps1$^{570A}$-3xFLAG *hog1Δ* (yAM291-A) and Fps1$^{3A}$-3xFLAG *hog1Δ* (yAM278) strains were adjusted to A$_{600\ nm}$ = 1.0 and serial dilutions were then spotted onto YPD plates containing the indicated concentration of arsenite. Cells were allowed to grow for 2 days at 30°C prior to imaging. (**B**) As in (**A**), except Fps1$^{IVAA}$-3xFLAG (yAM308-A), Fps1$^{(3A)IVAA}$-3xFLAG (yAM309-A), Rgc2$^{7A}$-HA (yAM315) and Fps1$^{3A}$-3xFLAG Rgc2$^{7A}$-HA (yAM318) strains were tested. The Fps1$^{IVAA}$ mutation prevents Hog1 binding to and regulation of Fps1, and Rgc2$^{7A}$ cannot be displaced from Fps1 because it cannot be phosphorylated by Hog1; both mutations render the channel constitutively open and make cells arsenite sensitive (*Lee et al., 2013*). (**C**) Fps1-3xFLAG (yAM271-A) or Fps1$^{3A}$-3xFLAG (yAM272-A) strains were co-transformed with P$_{MET25}$-Rgc2-HA (p3151) and P$_{MET25}$-Fps1-3xFLAG (pAX302) or P$_{MET25}$-Fps1$^{3A}$ -3xFLAG (pAX303) plasmids. After Rgc2-HA and Fps1-3xFLAG expression, Fps1 was immuno-purified with anti-FLAG antibody-coated beads (see 'Materials and methods'). The bound proteins were resolved by SDS-PAGE and the amount of Rgc2-HA present determined by immunoblotting with anti-HA antibody. (**D**) Wild-type (BY4741), *hog1Δ* (YJP544) or Fps1$^{3A}$-3xFLAG *hog1Δ* (yAM278) strains were grown and serial dilutions of these cultures plated onto synthetic complete medium lacking tryptophan with 2% dextrose and the indicated concentration of sorbitol. Cells were grown for 3 days prior to imaging.

Collectively, our results show that, independently of Hog1, hypertonic conditions drastically diminish TORC2-dependent Ypk1 phosphorylation, in turn dramatically decreasing Ypk1-mediated Fps1 phosphorylation, thereby closing the channel and causing intracellular glycerol accumulation. Thus, absence of Ypk1 phosphorylation should allow a cell lacking Hog1 to better survive hyperosmotic conditions. Indeed, Fps1[3A] hog1Δ cells are significantly more resistant to hyperosmotic stress than otherwise isogenic hog1Δ cells (*Figure 3D*). This epistasis confirms that, even when Hog1 is absent, loss of Ypk1-mediated Fps1 channel opening is sufficient for cells to accumulate an adequate amount of glycerol to physiologically cope with hyperosmotic stress.

## Discussion

Aside from further validating the utility of our screen for identifying new Ypk1 substrates (*Muir et al., 2014*), our current findings demonstrate that TORC2-dependent Ypk1-catalyzed phosphorylation of Fps1 opens this channel and, conversely, that loss of Ypk1-dependent Fps1 phosphorylation upon hypertonic shock is sufficient to close the channel, prevent glycerol efflux, and promote cell survival. In agreement with our observations, in a detailed kinetic analysis of global changes in the *S. cerevisiae* phosphoproteome upon hyperosmotic stress (*Kanshin et al., 2015*), it was noted that two sites in Fps1 (S181 and T185), which we showed here are modified by Ypk1, become dephosphorylated.

We previously showed that Gpd1, the rate-limiting enzyme for glycerol production under hyperosmotic conditions (*Remize et al., 2001*), is negatively regulated by Ypk1 phosphorylation (*Lee et al., 2012*). Thus, inactivation of TORC2-Ypk1 signaling upon hyperosmotic shock has at least two coordinated consequences that work synergistically to cause glycerol accumulation and promote cell survival, a similar outcome but mechanistically distinct from the processes evoked by Hog1 activation (*Figure 4*). First, loss of TORC2-Ypk1 signaling alleviates inhibition of Gpd1, which, combined with transcriptional induction of *GPD1* by hyperosmotic stress, greatly increases glycerol production. Second, loss of TORC2-Ypk1 signaling closes the Fps1 channel, thereby retaining the glycerol produced.

Presence of two systems (TORC2-Ypk1 and Hog1) might allow cells to adjust optimally to stresses occurring with different intensity, duration, or frequency. Reportedly, Hog1 responds to stresses occurring no more frequently than every 200 s (*Hersen et al., 2008*; *McClean et al., 2009*), whereas we found TORC2-Ypk1 signaling responded to hypertonic stress in ≤60 s. Also, the Sln1 and Sho1 sensors that lead to Hog1 activation likely can respond to stimuli that do not affect the TORC2-Ypk1 axis, and vice-versa.

A remaining question is how hyperosmotic stress causes such a rapid and profound reduction in phosphorylation of Ypk1 at its TORC2 sites. This outcome could arise from activation of a phosphatase (other than CN), inhibition of TORC2 catalytic activity, or both. Despite a recent report that Tor2 (the catalytic component of TORC2) interacts physically with Sho1 (*Lam et al., 2015*), raising the possibility that a Hog1 pathway sensor directly modulates TORC2 activity, we found that hyperosmolarity inactivates TORC2 just as robustly in *sho1Δ* cells as in wild-type cells. Alternatively, given the role ascribed to the ancillary TORC2 subunits Slm1 and Slm2 (*Gaubitz et al., 2015*) in delivering Ypk1 to the TORC2 complex (*Berchtold et al., 2012*; *Niles et al., 2012*), response to hyperosmotic shock might be mediated by some influence on Slm1 and Slm2. Thus, although the mechanism that abrogates TORC2 phosphorylation of Ypk1 upon hypertonic stress remains to be delineated, this effect and its consequences represent a novel mechanism for sensing and responding to hyperosmolarity.

## Materials and methods

### Construction of yeast strains and growth conditions

*S. cerevisiae* strains used in this study (*Supplementary file 1*) were constructed using standard yeast genetic manipulations (*Amberg et al., 2005*). For all strains constructed, integration of each DNA fragment of interest into the correct genomic locus was assessed using genomic DNA from isolated colonies of corresponding transformants as the template and PCR amplification with an oligonucleotide primer complementary to the integrated DNA and a reverse oligonucleotide primer complementary to chromosomal DNA at least 150 bp away from the integration site, thereby confirming that the DNA fragment was integrated at the correct locus. Finally, the nucleotide sequence of each resulting reaction product was determined to confirm that it had the correct

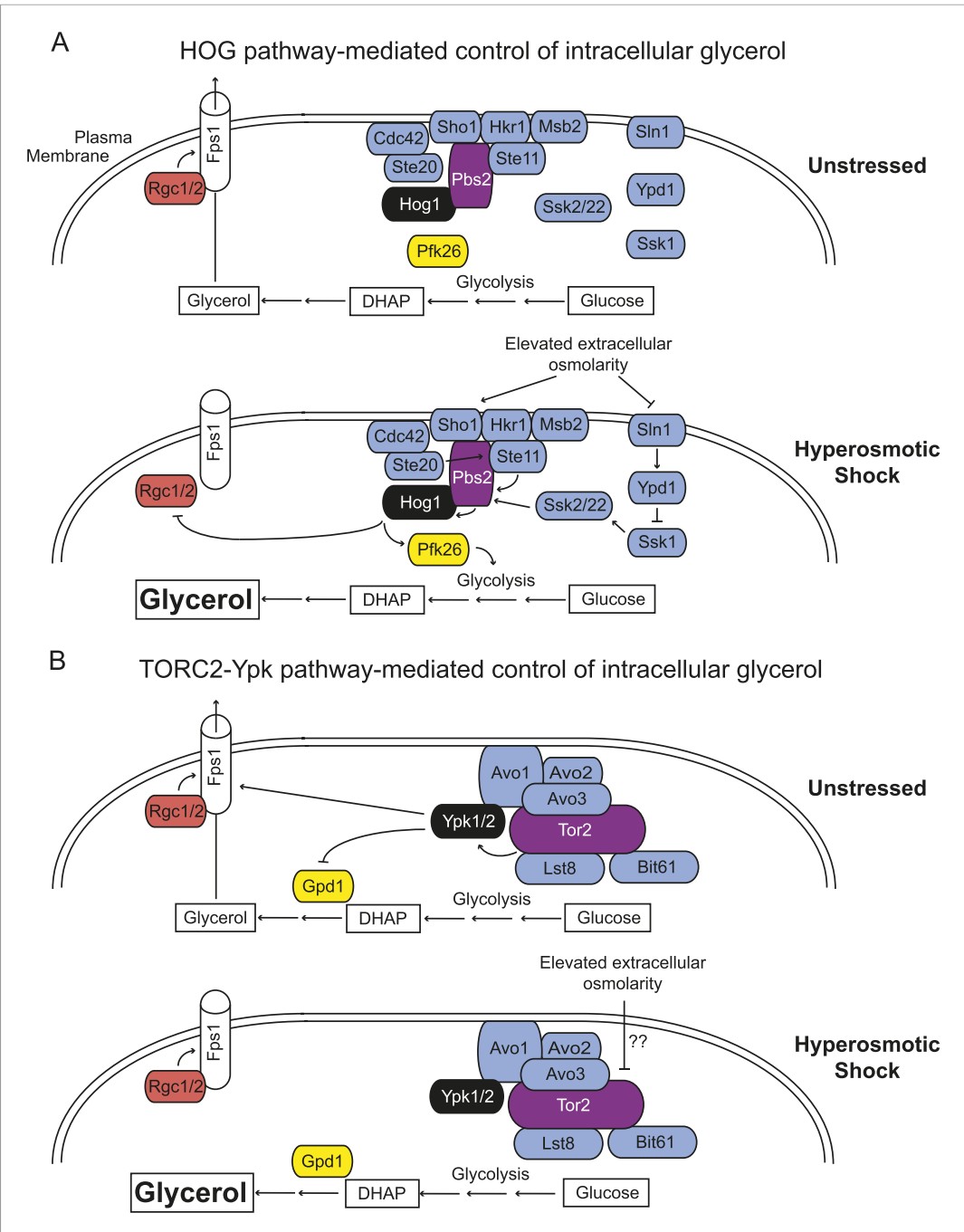

**Figure 4**. *Saccharomyces cerevisiae* has two independent sensing systems to rapidly increase intracellular glycerol upon hyperosmotic stress. (**A**) Hog1 MAPK-mediated response to acute hyperosmotic stress (adapted from ***Hohmann, 2015***). Unstressed condition (*top*), Hog1 is inactive and glycerol generated as a minor side product of glycolysis under fermentation conditions can escape to the medium through the Fps1 channel maintained in its open state by bound Rgc1 and Rgc2. Upon hyperosmotic shock (*bottom*), pathways coupled to the Sho1 and Sln1 osmosensors lead to Hog1 activation. Activated Hog1 increases glycolytic flux via phosphorylation of Pkf26 in the cytosol and, on a longer time scale, also enters the nucleus (not depicted) where it transcriptionally upregulates *GPD1* (***de Nadal et al., 2011***; ***Saito and Posas, 2012***), the enzyme rate-limiting for glycerol formation, thereby increasing glycerol production. Activated Hog1 also prevents glycerol efflux by phosphorylating and displacing the Fps1 activators Rgc1 and Rgc2 (***Lee et al., 2013***). These processes act synergistically to elevate the intracellular glycerol concentration providing an osmolyte to counterbalance the external high osmolarity. (**B**) Unstressed condition (*top*), active TORC2-Ypk1 keeps intracellular glycerol level low by inhibition of Gpd1 (***Lee et al., 2012***) and *Figure 4. continued on next page*

*Figure 4. Continued*

because Ypk1-mediated phosphorylation promotes the open state of the Fps1 channel. Upon hyperosmotic shock (*bottom*), TORC2-dependent phosphorylation of Ypk1 is rapidly down-regulated. In the absence of Ypk1-mediated phosphorylation, inhibition of Gpd1 is alleviated, thereby increasing glycerol production. Concomitantly, loss of Ypk1-mediated phosphorylation closes the Fps1 channel, even in the presence of Rgc1 and Rgc2, thereby promoting glycerol accumulation to counterbalance the external high osmolarity. Schematic depiction of TORC2 based on data from *Wullschleger et al. (2005)*; *Liao and Chen (2012)*; *Gaubitz et al. (2015)*.

sequence. Yeast cultures were grown in rich medium (YPD; 1% yeast extract, 2% peptone, 2% glucose) or in defined minimal medium (SCD; 0.67% yeast nitrogen base, 2% glucose) supplemented with the appropriate nutrients to permit growth of auxotrophs and/or to select for plasmids.

## Plasmids and recombinant DNA methods

All plasmids used in this study (*Supplementary file 2*) were constructed using standard laboratory methods (*Green and Sambrook, 2012*) or by Gibson assembly (*Gibson et al., 2009*) using the Gibson Assembly Master Mix Kit according to the manufacturer's specifications (New England Biolabs, Ipswich, Massachusetts, United States). All constructs generated in this study were confirmed by nucleotide sequence analysis covering all promoter and coding regions in the construct.

## Preparation of cell extracts and immunoblotting

Yeast cell extracts were prepared by an alkaline lysis and trichloroacetic acid (TCA) precipitation method, as described previously (*Westfall et al., 2008*). For samples analyzed by immunoblotting, the precipitated proteins were resolubilized and resolved by SDS-PAGE, as described below. For samples subjected to phosphatase treatment, the precipitated proteins were resolubilized in 100 μl solubilization buffer (2% SDS, 2% β-mercaptoethanol, 150 mM NaCl, 50 mM Tris-HCl [pH 8.0]), diluted with 900 μl calf intestinal phosphatase dilution buffer (11.1 mM $MgCl_2$, 150 mM NaCl, 50 mM Tris-HCl [pH 8.0]), incubated with calf intestinal alkaline phosphatase (350 U; New England Biolabs) for 4 hr at 37°C, recollected by TCA precipitation, resolved by SDS-PAGE, and analyzed by immunobotting. To resolve Gpt2 and its phosphorylated isoforms, samples (15 μl) of solubilized protein were subjected to SDS-PAGE at 120 V in 8% acrylamide gels polymerized and crosslinked with a ratio of acrylamide:bisacrylamide::75:1. To resolve Fps1 and Ypk1 and their phosphorylated isoforms, samples (15 μl) of solubilized protein were subjected to Phos-tag SDS-PAGE (*Kinoshita et al., 2009*) (8% acrylamide, 35 μM Phos-tag [Wako Chemicals USA, Inc.], 35 μM $MnCl_2$) at 160 V.

After SDS-PAGE, proteins were transferred to nitrocellulose and incubated with mouse or rabbit primary antibody in Odyssey buffer (Li-Cor Biosciences, Lincoln, Nebraska, United States), washed, and incubated with appropriate IRDye680LT-conjugated or IRDye800CW-conjugated anti-mouse or anti-rabbit IgG (Li-Cor Biosciences) in Odyssey buffer with 0.1% Tween-20 and 0.02% SDS. Blots were imaged using an Odyssey infrared scanner (Li-Cor Biosciences). Primary antibodies and dilutions used were: rabbit anti-HA, 1:1000 (Covance Inc., Dedham, Massachusetts, United States); mouse anti-HA, 1:1000 (Covance Inc.); mouse anti-FLAG, 1:5000 (Sigma–Aldrich, St. Louis, Missouri, United States); rabbit anti-FLAG, 1:5000 (Sigma–Aldrich); tissue culture medium containing mouse anti-c-myc mAb 9E10, 1:100 (Monoclonal Antibody Facility, Cancer Research Laboratory, University of California, Berkeley); rabbit anti-Ypk1(P-T662), 1:20,000 (generous gift from Ted Powers, University of California, Davis); and, rabbit anti-yeast Pgk1, 1:10,000 (this laboratory).

## Protein purification and in vitro kinase assay

Ypk1 and GST-Fps1(531-0669) proteins were purified as previously described (*Muir et al., 2014*). Following protein purification, Ypk1 in vitro kinase assays were performed as previously described (*Muir et al., 2014*).

## Measurement of intracellular glycerol accumulation

Measurement of intracellular glycerol was conducted as described (*Albertyn et al., 1994a*). Briefly, samples (~40 ml) of exponentially-growing cultures were harvested by centrifugation, washed with 1 ml of medium, recollected and the resulting cell pellets frozen in liquid $N_2$ and stored at −80°C prior

to analysis. Each cell pellet was boiled for 10 min in 1 ml of 50 mM Tris-Cl (pH 7.0). This eluate was clarified by centrifugation for 15 min at 13,200 rpm (16,100×$g$) in a microfuge (Eppendorf 5415D). Glycerol concentration in the resulting supernatant fraction was measured using a commercial enzymic assay kit (Sigma Aldrich) and normalized to the protein concentration of the same initial extract as measured by the Bradford method (*Bradford, 1976*).

## Fluorescence microscopy of Fps1-GFP

An *fps1Δ* strain was transformed with plasmids expressing wild-type Fps1-GFP or the mutant Fps1-GFP derivatives and grown in selective medium to mid-exponential phase. Samples of the resulting cultures were viewed directly under an epifluorescence microscope (model BH-2; Olympus America, Inc.) using a 100× objective fitted with appropriate band-pass filters (Chroma Technology Corp.). Images were collected using a CoolSNAP MYO charge-coupled device camera (Photometrics, Tucson, Arizona, United States).

## Co-immunoprecipitation of Fps1 and Rgc2

Co-immunoprecipitation experiments were performed with minor modifications as previously described (*Lee et al., 2013*). Cells expressing Fps1-3xFLAG (yAM271-A), Fps1$^{3A}$-3xFLAG (yAM272-A) or untagged Fps1 (BY4742) were transformed with empty vector or the same vector expressing Fps1-3xFLAG (pAX302) or Fps1$^{3A}$-3xFLAG (pAX303) under control of the *MET25* promoter. These transformants were then co-transformed with a plasmid expressing Rgc2-3xHA under control of the *MET25* promoter (*Lee et al., 2013*). Cultures of each were grown to mid-exponential phase in SCD-Ura-Leu. Cultures were then diluted to A$_{600}$$_{nm}$ = 0.2 in 1 l of SCD-Ura-Leu-Met to induce expression of Rgc2-3xHA and Fps1-3xFLAG and grown at 30°C for 4 hr. Cells were harvested by centrifugation and resuspended in 5 ml of TNE+Triton+NP-40 (50 mM Tris-Cl [pH 7.5], 150 mM NaCl, 4 mM NaVO$_4$, 50 mM NaF, 20 mM Na-PPi, 5 mM EDTA, 5 mM EGTA, 0.5% Triton-X100, 1.0% NP-40, 1× cOmplete protease inhibitor [Roche, Pleasanton, California, United States]). The cells were then lysed cryogenically using Mixer Mill MM301 (Retsch GmbH, Haan, Germany). The lysate was thawed on ice and then clarified by centrifugation for 20 min at 10,500 rpm (13,000×$g$) in the SS34 rotor of a refrigerated centrifuge (Sorvall RC-5B). Protein concentration of the clarified lysate was measured using BCA reagent (Thermo Fisher Scientific, Waltham, Massachusetts, United States) and then Fps1-3xFLAG was immunoprecipitated from a volume of extract containing a total of 10 mg protein using 50 µl of mouse anti-FLAG antibody coupled-agarose resin (Sigma Aldrich) equilibrated in TNE+Triton+NP-40. Binding was allowed to occur for 2 hr at 4°C. The resin was then washed extensively with TNE+Triton+NP-40 and the proteins remaining bound were then resolved by SDS-PAGE and analyzed by immunoblotting with appropriate antibodies to detect both Fps1-3xFLAG and Rgc2-3xHA.

## Acknowledgements

This work was supported by NIH Predoctoral Training Grant GM07232 and a Predoctoral Fellowship from the UC Systemwide Cancer Research Coordinating Committee (to AM), by NIH Predoctoral Training Grant GM07232 (to KLL), by NIH R01 Research Grant GM21841 and Senior Investigator Award 11-0118 from the American Asthma Foundation (to JT). We thank Stefan Hohmann (Univ. of Göteborg, Sweden), David E Levin (Boston Univ., Boston, MA), and Ted Powers (Univ. of California, Davis) for generously providing strains, plasmids and reagents, Hugo Tapia (Koshland Lab, UC Berkeley) for helpful discussions and reagents for measuring intracellular glycerol, and Jesse Patterson and the other members of the Thorner Lab for various research materials and thoughtful suggestions.

## Additional information

### Funding

| Funder | Grant reference | Author |
| --- | --- | --- |
| National Institute of General Medical Sciences (NIGMS) | T32 GM07232 | Alexander Muir, Kristin L Leskoske |
| University of California Berkeley (University of California, Berkeley) | Predoctoral Fellowship | Alexander Muir |

| Funder | Grant reference | Author |
|---|---|---|
| National Institute of General Medical Sciences (NIGMS) | R01 GM21841 | Jeremy Thorner |
| Foundation of the American College of Allergy, Asthma & Immunology (ACAAI Foundation) | Senior Investigator Award 11-0118 | Jeremy Thorner |

The funders had no role in study design, data collection and interpretation, or the decision to submit the work for publication.

### Author contributions

AM, FMR, Conception and design, Acquisition of data, Analysis and interpretation of data, Drafting or revising the article; GT, Conception and design, Acquisition of data, Drafting or revising the article; KLL, Acquisition of data, Drafting or revising the article; JT, Conception and design, Analysis and interpretation of data, Drafting or revising the article

## Additional files

### Supplementary files

• Supplementary file 1. Yeast strains used in this study.

• Supplementary file 2. Plasmids used in this study.

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
