## [Decision Letter]

Thank you for submitting your work entitled “Down-regulation of TORC2-Ypk1 signaling promotes MAPK-independent survival under hyperosmotic stress” for peer review at *eLife*. Your submission has been favorably evaluated by Tony Hunter (Senior editor) and two reviewers. One of the two reviewers, Susan Henry, has agreed to share her identity.

The reviewers have discussed the reviews with one another, and the Reviewing editor has drafted this decision to help you prepare a revised submission.

The reviewers were interested in your elucidation of a new TORC2-Ypk1-Fps1 pathway that serves as a Hog1-independent backup mechanism for cell survival in response to hyperosmotic stress acting by closing the Fps1 channel for glycerol efflux, resulting in glycerol accumulation and cell survival during hyperosmotic stress, and they are both in favor of publication in *eLife* with revision.

Please address the specific points raised by Reviewer 2, which all appear feasible to do within a short time frame.

Reviewer 1:

Prior work from this group had shown that phosphorylation of the Ypk1 protein kinase at its TORC2-dependent sites is eliminated rapidly during hyperosmotic shock but that this loss is attenuated within 10-15 min, but this attenuation was found to be independent of known stress response pathways, including the Hog1 MAPK. Their previous proteomic screen had also identified Fps1p as being a target of the Ypk1p protein kinase.

This current paper provides strong and clear evidence that the aquaglyceroporin Fps1p is a bona fide target of Ypk1, a TORC2-dependent protein kinase, upon exposure of yeast cells to hypertonic stress. This current work also demonstrates for the first time that the open channel state of Fps1 is dependent on phosphorylation by Ypk1p. In addition, and importantly, the authors show that in wild-type cells, under hyperosmotic stress conditions, TORC2-dependent phosphorylation of Fps1 by Ypk1 is blocked. They show that this results in closure of the Fps1 channel for glycerol efflux, resulting in glycerol accumulation during hyperosmotic stress, which in turn leads to increased cell survival under these conditions.

The evidence presented also indicates that these responses are not dependent on the Hog1 MAPK pathway, which is also activated during hypertonic stress. Thus, this work identifies a novel mechanism involved in survival under hypertonic stress that is Hog1 independent, but TORC2-dependent.

Overall, the data are of high quality and importance and reveal a novel mechanism.

Reviewer 2:

Muir et al. follow up a previous genome-wide screen in which they identified potential Ypk1 substrates. They now confirm that aquaglyceroporin (Fps1) is a target of Ypk1. The authors demonstrate that TORC2-Ypk1 signaling initiates Fps1 phosphorylation to promote Fps1 channel opening. Phospho-deficient Fps1 is in a closed state and confers resistance to hyperosmotic stress and arsenite. They also show that hyperosmotic stress inhibits TORC2-Ypk1 signaling.

1) It has been reported already that the Slm proteins activate TORC2-Ypk1 signaling in response to hypo-osmotic stress (Berchtold et al. Nat Cell Biol. 14:542, 2012). Do the Slm proteins inhibit TORC2-Ypk1 in response to hyperosmotic stress? Also, the Berchtold et al. studies should be discussed.

2) Figure 1: The authors show that TORC2-Ypk1 signaling promotes phosphorylation of Fps1 on three serine residues (Ser181, Ser 185, Ser570) as determined by resolving “phospho-”Fps1 in phos-tag gels. The phos-tag gel assay requires a phosphatase-treated control to claim that that the slow-migrating band is indeed phospho-Fps1. Figure 1 (experiment for Gpt2 phosphorylation) also needs a phosphatase-treated control.

3) Figure 1: The authors demonstrate that acute inactivation of TORC2, by BEZ235 treatment of an analog-sensitive *tor2* (*tor2-as*) mutant, reduces Fps1 phosphorylation. Is there a positive control to show that TORC2 activity is indeed inhibited? The authors could possibly look at TORC1 activity (e.g. Sch9 phosphorylation), since TOR2 is also part of TORC1.

4) Figure 1: The authors show that TORC2 activity is inhibited upon sorbitol treatment, as determined by mobility shift of Ypk1 in a phos-tag gel. For this experiment, they used a phospho-deficient mutant of Ypk1 in which bona fide TORC2 target sites (Ser644 and Thr662) are mutated. This suggests that the slow migrating form of Ypk1 (this also needs phosphatase treatment) may be phosphorylated on yet to be characterized sites and it is these sites that the authors are monitoring. A more reliable assay of TORC2 activity would be to blot with phospho-Ypk1 antibody (Thr662) (Berchtold et al. Nat Cell Biol. Op. cit.).

5) The authors state: “Thus, Fps1 is a bona fide Ypk1 substrate.” To make this claim, they should provide evidence that Ypk1 directly phosphorylates Fps1. This can be done by performing a Ypk1 kinase assay in vitro with recombinant Fps1-WT and Fps1-3A. The authors have previously performed in vitro Ypk1 assays.

6) Figure 2: The authors show that phospho-deficient Fps1-3A cells are the most arsenite resistant cells among *fps1* mutants tested. They concluded that the Fps1^3A^ mutant is a closed channel such that toxic arsenite cannot enter cells. Why are Fps1^3A^ mutant cells more resistant than *fps1∆* cells or Fps1-∆PHD cells in which Fps1 channel is constantly absent or closed, respectively?

7) Figure 2: Based on data showing that Fps1^3A^ mutant cells are resistant to arsenite, the authors conclude that Ypk1-mediated phosphorylation promotes Fps1 channel opening. This conclusion would be strengthened if the authors also demonstrated that a phospho-mimetic mutant of Fps1 (Fps1^3D^ or Fps1^3E^) confers arsenite hyper-sensitivity.

---

## [Author Response]

*The reviewers were interested in your elucidation of a new TORC2-Ypk1-Fps1 pathway that serves as a Hog1-independent backup mechanism for cell survival in response to hyperosmotic stress acting by closing the Fps1 channel for glycerol efflux, resulting in glycerol accumulation and cell survival during hyperosmotic stress, and they are both in favor of publication in* eLife *with revision*.

*Please address the specific points raised by Reviewer 2, which all appear feasible to do within a short time frame*.

We are very pleased that the editor and the two referees are in favor of publication of our Research Advance in *eLife*, pending appropriate revision in response to the remarks of Reviewer #2. In this regard, and as enumerated below, several of the issues raised by Reviewer #2 are already addressed in prior published work from ourselves and others, and certain other issues are addressed by results now provided in the revised manuscript (which were not originally included in the initially submitted manuscript because of the length limitations of the Research Advance format). Some of the newly added data were generated by graduate student Kristin L. Leskoske and, hence, she has now been added as a co-author of this study.

Reviewer 2:

*Muir et al. follow up a previous genome-wide screen in which they identified potential Ypk1 substrates. They now confirm that aquaglyceroporin (Fps1) is a target of Ypk1. The authors demonstrate that TORC2-Ypk1 signaling initiates Fps1 phosphorylation to promote Fps1 channel opening. Phospho-deficient Fps1 is in a closed state and confers resistance to hyperosmotic stress and arsenite. They also show that hyperosmotic stress inhibits TORC2-Ypk1 signaling*.

We thank this referee for the generally correct synopsis of the novel findings reported in our study. However, this referee felt that we needed to better address and/or experimentally document several of our findings. Our disposition of those concerns is described below.

*1) It has been reported already that the Slm proteins activate TORC2-Ypk1 signaling in response to hypo-osmotic stress (Berchtold et al. Nat Cell Biol. 14:542, 2012). Do the Slm proteins inhibit TORC2-Ypk1 in response to hyperosmotic stress? Also, the Berchtold et al. studies should be discussed*.

The main focus of this Research Advance was to demonstrate unequivocally that Fps1 is indeed an authentic in vivo substrate of the TORC2-Ypk1 signaling axis and to elucidate the physiologically role of these post -translational modifications. Moreover, given the space limitations of a Research Advance, we feel that it is well beyond the scope of this report to demand that we also elucidate the mechanism by which hyperosmotic shock ablates transiently the TORC2-dependent phosphorylation of Ypk1, and whether the Slm1 and Slm2 proteins may or may not be involved at this level of regulation.

Nonetheless, in this same regard, existing data in the literature indicate that it would be technically very difficult to directly address the question posed by Reviewer #2 about whether the Slm1 (and/or Slm2) proteins are required for the observed down-regulation of TORC2-Ypk1 signaling under hyperosmotic conditions, for the following reasons. There are only two known conditions under which *slm1∆slm2∆* cells are viable and in which, *theoretically*, we could ask whether or not 1 M sorbitol is still able to cause a drop in TORC2-mediated phosphorylation of Ypk1 when the Slm1 and Slm2 proteins are absent.

First, a *slm1∆slm2∆sac7∆* triple mutant is viable, as shown by work from Scott Emr's lab [Audhya A et al. (2004) EMBO J*.* 23: 3747 -3757]. *SAC7* encodes a primary GAP for yeast Rho1 and, as shown in collaborative work by Yoshi Ohya and David Levin [Kamada Y et al. (1996) J. Biol. Chem. 271: 9193-9196], Rho1 is the activator of yeast Pkc1 [misnamed because it really is more related to the mammalian Rho- (and Rac-) activated PKR2/PKNγ; see Vincent S, Settleman J (1997) Mol. Cell. Biol*.* 17: 2247-2256; Mukai H (2003*) J. Biochem. (Tokyo)* 133: 17-27]. In any event, loss of Sac7 up-regulates Pkc1 signaling and, as we demonstrated before [Roelants FM et al*.* (2002) Mol. Biol. Cell 13: 3005 -3028], any perturbation that up-regulates Pkc1 function bypasses the inviability of cells deficient in Ypk1 and Ypk2 signaling, presumably because some aspects of the cell wall integrity pathway act in parallel or are semi-redundant with functions normally controlled by Ypk1 and Ypk2. The problem arises, however, from observations reported by Ted Powers lab [Niles BJ et al. (2012) PNAS 109: 1536–1541]. They examined the state of Ypk1 phosphorylation in *slm1∆slm2∆sac7∆* triple mutant and found that Ypk1 phosphorylation at both its activation loop (T504) and at one of its critical TORC2 sites (T662), as judged for the latter using a phospho-site-specific antibody they themselves generated, were greatly reduced. Thus, because TORC2- mediated phosphorylation is already way down in this background, it would be very challenging and probably not meaningful to examine whether the level of T662 phosphorylation goes down further in response to challenge with 1 M sorbitol.

Second, a *slm1∆slm2∆* double mutant expressing a constitutively-active allele, Ypk1(D242A), is also able to grow. However, this allele is able to bypass the need for functional TORC2, as judged by its ability to rescue the lethality of a *tor2*^*ts*^ mutant or treatment of cells with a TORC2 inhibitor [Roelants FM et al. (2011) PNAS 108: 19222-19227] or *avo3∆* cells, which are deficient in TORC2 activity [Niles, op. cit.]. However, the same problem arises here too because, as judged by using exactly the same antibody (generously provided by Ted Powers), we have found that T662 phosphorylation is virtually undetectable on Ypk1(D242A) in *slm1∆slm2∆* cells (K. Leskoske, unpublished data). Thus, once again, because TORC2-mediated phosphorylation is already way down in this background, it would be very challenging and probably not meaningful to examine whether the level of T662 phosphorylation goes down further in response to challenge with 1 M sorbitol.

Finally, although the lethality of Slm1- and Slm2-deficient cells clearly arises from a lack of sufficient Ypk1 function, exactly why Slm1- and Slm2-deficient cells have low apparent TORC2-mediated phosphorylation of Ypk1 is not clear. Slm1 and Slm2 were identified originally as PP2B/calcineurin-binding proteins nearly contemporaneously in three different labs [Bultynck G et al. (2006) Mol. Cell. Biol. 26: 4729-4745; Tabuchi M et al. (2006) Mol. Cell. Biol*.* 26 : 5861 -5875; Daquinag A et al. (2007) Mol. Cell. Biol*.* 27: 633-650]. Hence, in the absence of Slm1 and Slm2, there will be a lack of proper sequestration of PP2B. Thus, the low level of Ypk1 phosphorylation observed when Slm1 and Slm2 are absent could arise, potentially, from an enhanced rate of dephosphorylation of Ypk1 itself or of some site in a component of TORC2 critical for its activity on Ypk1, or both. Alternatively, because Slm1 and Slm2 contain lipid-binding PH domains near their C-terminus [Yu JW et al. (2004) Mol. Cell 13 : 677-688; Gallego O et al. (2010) Mol. Syst. Biol*.* 6: 430.1-430.15] and both are considered subunits of TORC2 [Gaubitz C et al. (2015) Mol. Cell 58: 977- 988] it is possible that, in the absence of Slm1 and Slm2, inefficient or improper membrane tethering of TORC2 is responsible for the reduction in Ypk1 phosphorylation observed in *slm1∆slm2∆* cells. Neither we nor any other yeast researcher has yet adequately resolved these issues. Nonetheless, as requested by Reviewer #2, we have added a sentence to the Discussion pointing out that it is possible that the response to hyperosmotic shock might be mediated by some influence on the Slm1 and Slm2 proteins, and we cite the Berchtold et al. reference in this context (although the Berchtold paper was already cited in our original manuscript).

*2)*
Figure 1*: The authors show that TORC2-Ypk1 signaling promotes phosphorylation of Fps1 on three serine residues (Ser181, Ser 185, Ser570) as determined by resolving “phospho-”Fps1 in phos-tag gels. The phos-tag gel assay requires a phosphatase-treated control to claim that that the slow-migrating band is indeed phospho-Fps1.*
Figure 1
*(experiment for Gpt2 phosphorylation) also needs a phosphatase-treated control*.

We showed, using Phos-tag™ gels, that the migration of Fps1^3A^ in vivo exactly mirrors the collapse of isoforms of wild-type Fps1 observed in vivo upon inhibition of Ypk1 function. Thus, the inescapable conclusion is that Fps1 is indeed phosphorylated at these three sites in the cell and that Ypk1 is the enzyme responsible for these phosphorylations in vivo. Nonetheless, as requested by Reviewer #2, we now provide new data (new Figure 1—figure supplement 2) that directly address this issue by showing that CIP treatment of WT Fps1 collapses the isoforms to a migration pattern identical to Fps1^3A^. As also requested by Reviewer #2, the same analysis is now also provided for Gpt2 (new Figure 1—figure supplement 1), even though it is not the primary focus of the findings presented in this paper.

*3)*
Figure 1*: The authors demonstrate that acute inactivation of TORC2, by BEZ235 treatment of an analog-sensitive* tor2 *(*tor2-as*) mutant, reduces Fps1 phosphorylation. Is there a positive control to show that TORC2 activity is indeed inhibited? The authors could possibly look at TORC1 activity (e.g. Sch9 phosphorylation), since TOR2 is also part of TORC1*.

Data fully documenting the efficacy and specificity of the yeast *tor2-as* allele for inhibition by NVP-BEZ235 were already published by the lab of Kevan Shokat [Kliegman JI et al. (2013) Cell Rep. 5: 1725-173]. We see no necessity to recapitulate those experiments, especially because our results indicate that the inhibitor reduces phosphorylation of Fps1 at its Ypk1 sites, as expected because Ypk1 activity is TORC2-dependent, fully consistent with the conclusions of Kliegman [op. cit.].

*4)*
Figure 1*: The authors show that TORC2 activity is inhibited upon sorbitol treatment, as determined by mobility shift of Ypk1 in a phos-tag gel. For this experiment, they used a phospho-deficient mutant of Ypk1 in which bona fide TORC2 target sites (Ser644 and Thr662) are mutated. This suggests that the slow migrating form of Ypk1 (this also needs phosphatase treatment) may be phosphorylated on yet to be characterized sites and it is these sites that the authors are monitoring. A more reliable assay of TORC2 activity would be to blot with phospho-Ypk1 antibody (Thr662) (Berchtold et al. Nat Cell Biol. Op. cit.)*.

As will be described in detail in another manuscript (Leskoske K, Roelants FM and Thorner J), TORC2 does indeed phosphorylate Ypk1 both in vivo and in vitro at three additional sites aside from its classical “turn” (S644) and “hydrophobic” (T662) motifs. So, the isoforms generated on Phos-tag™ gels by the Ypk1^7A^ mutant used are a reliable reporter of its TORC2-mediated modification. Nonetheless, as requested by Reviewer #2, we now provide new data (new Figure 1—figure supplement 4) that directly address this issue by showing using the reagent recommended by the referee, namely anti-P-T662 site antibody (again, generously provided by Ted Powers), that phosphorylation at this site is rapidly and completely abrogated when cells are treated with 1 M sorbitol, in agreement with our independent analysis using Ypk1^7A^. In addition, we now demonstrate by phosphatase treatment that the observed isoforms are indeed due to phosphorylation (new Figure 1—figure supplement 4), as also requested by Reviewer #2.

*5) The authors state: “Thus, Fps1 is a bona fide Ypk1 substrate.” To make this claim, they should provide evidence that Ypk1 directly phosphorylates Fps1. This can be done by performing a Ypk1 kinase assay in vitro with recombinant Fps1-WT and Fps1-3A. The authors have previously performed in vitro Ypk1 assays*.

Like other demonstrated targets of Ypk1 (e.g. Orm1, Orm2, Lac1, Lag1), Fps1 is a polytopic integral membrane protein. Hence, as for these other substrates, we use large soluble fragments of the protein that contain its cytosolically disposed domains as the phospho-acceptor in in vitro kinase reactions. Thus, as requested by Reviewer #2, and as an illustrative example, we now provide new data (new Figure 1—figure supplement 3) that directly address this issue by documenting that a large (139-residue) fragment of Fps1 containing one of the three primary Ypk1 phosphorylation sites observed in vivo (S570) is indeed phosphorylated in Ypk1-specific manner in vitro and solely at this Ypk1 site (S570), as we previously described as “data not shown” [Muir A et al. (2014) *eLife*
**3**: e03779.1-e03779.34].

*6)*
Figure 2*: The authors show that phospho-deficient Fps1-3A cells are the most arsenite resistant cells among* fps1 *mutants tested. They concluded that the Fps1*^*3A*^
*mutant is a closed channel such that toxic arsenite cannot enter cells. Why are Fps1*^*3A*^
*mutant cells more resistant than* fps1∆ *cells or Fps1-∆PHD cells in which Fps1 channel is constantly absent or closed*, *respectively?*

The particular image provided to illustrate that the Fps1^3A^ mutant confers arsenite resistance was included because, as Reviewer #2 noted, it contained a full set of controls. Moreover, we have done the comparison multiple times and Fps1^3A^-expressing cells do grow slightly better than cells totally lacking Fps1 (*fps1∆* mutant) on *both* YPD alone and YPD + arsenite. The modest, but measurable, decrease in vegetative growth rate caused by the complete absence of Fps1 has been noted previously by others [Yoshikawa K et al. (2011) Yeast 28: 349-361; Lourenco AB et al. (2013) PLoS One 8: e55439.1-e55439.12; Marek A, Korona R (2013) Evolution 67 : 3077-3086]. Presumably, retention of this channel [∼1,000 copies per cell; Ghaemmaghami S et al. (2003) Nature 425: 737-741], even in closed form, preserves some aspect of cell function that becomes slightly compromised when this integral plasma membrane protein is missing.

*7)*
Figure 2*: Based on data showing that Fps1*^*3A*^
*mutant cells are resistant to arsenite, the authors conclude that Ypk1-mediated phosphorylation promotes Fps1 channel opening. This conclusion would be strengthened if the authors also demonstrated that a phospho-mimetic mutant of Fps1 (Fps1*^*3D*^
*or Fps1*^*3E*^*) confers arsenite hyper-sensitivity*.

We tested, of course, whether an Fps1^3E^ mutant, which could possibly mimic the “persistently phosphorylated” and thus the “permanently open state” of the channel, might confer a degree of arsenite sensitivity equivalent to or even greater than that conferred by WT Fps1. In this instance, however, we found that an Fps1^3E^ mutant exhibited arsenite resistance, comparable to that displayed by Fps1^3A^. Therefore, it seems that, in the case of this particular protein and these particular sites, Ser-to-Glu mutations are not an adequate mimic for authentic phosphate groups.